# Adipose Stromal Cell-Derived Secretome Attenuates Cisplatin-Induced Injury In Vitro Surpassing the Intricate Interplay between Proximal Tubular Epithelial Cells and Macrophages

**DOI:** 10.3390/cells13020121

**Published:** 2024-01-09

**Authors:** Erika Rendra, Stefanie Uhlig, Isabell Moskal, Corinna Thielemann, Harald Klüter, Karen Bieback

**Affiliations:** 1Institute of Transfusion Medicine and Immunology, Medical Faculty Mannheim, Heidelberg University, German Red Cross Blood Service Baden-Württemberg-Hessen, 68167 Mannheim, Germany; erika.erika@medma.uni-heidelberg.de (E.R.); harald.klueter@medma.uni-heidelberg.de (H.K.); 2Flow Core Mannheim, Medical Faculty Mannheim, Heidelberg University, 68167 Mannheim, Germany; 3Mannheim Institute for Innate Immunoscience, Medical Faculty Mannheim, Heidelberg University, 68167 Mannheim, Germany

**Keywords:** MSC secretome, proximal tubular epithelial cells, cisplatin, macrophages, acute kidney injury, oxidative stress, crosstalk, phagocytosis

## Abstract

(1) Background: The chemotherapeutic drug cisplatin exerts toxic side effects causing acute kidney injury. Mesenchymal stromal cells can ameliorate cisplatin-induced kidney injury. We hypothesize that the MSC secretome orchestrates the vicious cycle of injury and inflammation by acting on proximal tubule epithelial cells (PTECs) and macrophages individually, but further by counteracting their cellular crosstalk. (2) Methods: Conditioned medium (CM) from adipose stromal cells was used, first assessing its effect on cisplatin injury in PTECs. Second, the effects of cisplatin and the CM on macrophages were measured. Lastly, in an indirect co-culture system, the interplay between the two cell types was assessed. (3) Results: First, the CM rescued PTECs from cisplatin-induced apoptosis by reducing oxidative stress and expression of nephrotoxicity genes. Second, while cisplatin exerted only minor effects on macrophages, the CM skewed macrophage phenotypes to the anti-inflammatory M2-like phenotype and increased phagocytosis. Finally, in the co-culture system, the CM suppressed PTEC death by inhibiting apoptosis and nuclei fragmentation. The CM lowered TNF-α release, while cisplatin inhibited macrophage phagocytosis, PTECs, and the CM to a greater extent, thus enhancing it. The CM strongly dampened the inflammatory macrophage cytokine secretion triggered by PTECs. (4) Conclusions: ASC-CM surpasses the PTEC–macrophage crosstalk in cisplatin injury. The positive effects on reducing cisplatin cytotoxicity, on polarizing macrophages, and on fine-tuning cytokine secretion underscore MSCs’ CM benefit to prevent kidney injury progression.

## 1. Introduction

Mesenchymal stromal cells (MSCs) are considered promising candidates for the development of cell-based therapies for a variety of diseases [1]. The high interest in exploiting MSCs’ clinical potential stems from their pro-angiogenic, anti-apoptotic, anti-oxidative, and immunomodulatory properties [2,3]. The underlying mechanism of action by which MSCs deliver those therapeutic benefits includes the release of bioactive paracrine factors, collectively regarded as the secretome [4]. In general, the MSC secretome consists of soluble factors, which include various growth factors, cytokines, hormones, chemokines, anti-oxidants, extracellular matrix proteins, and other serum proteins, and extracellular vesicles (EVs). These EVs deliver proteins, lipids, and genetic materials (DNA, mRNA, and miRNA) to exert their effect on target cells. Given its diverse contents, the MSC secretome presents an option for multimodal treatment for debilitating diseases, such as kidney injury [1].

Cis-diamminedichloroplatinum II (cisplatin) is an anti-cancer drug with a wide range of side effects, including gastrotoxicity, neurotoxicity, allergic reactions, and, most prevalent of all, nephrotoxicity [5]. Acute kidney injury (AKI) happens to 20–30% of cancer patients receiving cisplatin, and it accounts for up to 60% of hospital-acquired kidney injuries with high mortality and morbidity [6]. Because of this, cisplatin-induced AKI has been the dose-limiting factor of cisplatin treatment, significantly compromising its efficacy in treating cancer [6,7,8].

Proximal tubular epithelial cells (PTECs) are the key player in cisplatin-induced AKI. Being responsible for reabsorption, PTECs express organic cation transporter 2 (OCT2), multidrug and toxin extrusion (MATE), and human copper transporter 1 (Ctr1), enabling cisplatin uptake [8,9,10]. Once internalized, cisplatin damages mitochondria by targeting mitochondrial DNA and the electron transport chain, resulting in increased reactive oxygen species (ROS) levels, the release of cytochrome C, and subsequent activation of the intrinsic apoptosis pathway [6,7]. Furthermore, cisplatin-induced ROS drive inflammation by the stimulation of the p38-MAPK pathway and the release of death receptor ligands, including tumor necrosis factor-α (TNF-α). The secretion of TNF-α further promotes PTEC apoptosis through the extrinsic pathway, which consecutively exacerbates intracellular ROS production [6,11].

Previous studies have demonstrated that the interplay between renal cells and macrophages is essential for determining the fate of AKI progression and recovery [12]. Soon after the insult, tubular cells, in particular PTECs, release chemotactic factors comprising pro-inflammatory cytokines and danger-associated molecular patterns (DAMPs) that promote macrophage infiltration and induce polarization into the pro-inflammatory phenotype, M1 [13]. Upon healing, M1 macrophages become gradually replaced by M2 macrophages, contributing to inflammation resolution and tubule regeneration [12,13,14]. However, when the injury is sustained, PTECs maintain the molecular pathways which favor M1 polarization, resulting in prolonged inflammation and further renal deterioration [14,15]. Kidney regeneration has been shown to require active resolution of renal inflammation by M2-like macrophages, highlighting their importance [16].

The role of macrophages in AKI has been largely studied in ischemia–reperfusion injury (IRI) models [17]. Yet the involvement of macrophages in cisplatin-induced AKI is not as extensively described [6,18,19,20]. Nakagawa et al. performed a phenotypical characterization of M1/M2 macrophages in cisplatin-induced renal fibrosis [21]. They showed that both CD68+ M1 and CD163+ M2 macrophages started infiltrating the kidney on day 5 post cisplatin treatment. While the M1 macrophage population peaked on day 9, the M2 population peaked later on day 12. Interestingly, on day 9, 60–80% of macrophages co-expressed M1 and M2 surface markers, eventually documenting the transition from M1 toward M2 macrophages. The authors concluded that a complicated interplay of inflammatory M1 and anti-inflammatory M2 macrophages was pivotal for cisplatin-induced renal fibrosis and its resolution.

Given the importance of PTECs’ role in macrophage polarization and vice versa [14,15], targeting the PTEC and macrophage crosstalk might pose a further additive effect in treating cisplatin-induced AKI. Indeed, given their anti-apoptotic and immunomodulatory capacity, MSCs have been shown to modulate both cell types individually [1,4]. We hypothesize that the MSCs can ameliorate cisplatin injury not only by directly rescuing PTECs and macrophages but also by having additive effects counteracting the vicious cycle of injury and inflammation. To address this, the direct effect of the MSC secretome on conditionally immortalized PTECs (ciPTECs) and macrophages upon cisplatin injury was evaluated first, separately, investigating potential mediators. Lastly, the MSC influence on PTEC and macrophage interaction was interrogated in a co-culture system, mainly by assessing the cytokine crosstalk.

## 2. Materials and Methods

### 2.1. MSC Culture and Production of Conditioned Medium (CM)

Adipose-derived mesenchymal stromal cells (ASCs) were isolated from the lipoaspirate of healthy donors upon obtaining informed consent in concordance with Mannheim Ethic Committee approval (ethical votes 2011-215N-MA, 2009-210N-MA), as previously described [22]. ASCs were cultivated in Dulbecco’s Modified Eagle Medium (DMEM; Lonza, #12-918F), supplemented with 10% (*v*/*v*) pooled human AB serum from healthy donors (German Red Cross Blood Donor Service, Mannheim, German), 1% Penicillin/Streptomycin (PAN Biotech, Aidenbach, Germany, # P06-07100), and 4 mM of L-glutamine (PAN Biotech, # P04-80100). ASCs were expanded, seeding 300 cells/cm^2^, and maintained in a cell culture incubator (37 °C, 5% CO_2_). ASCs were characterized with respect to their surface markers and osteogenic and adipogenic differentiation ability (Appendix A). Different ASC batches were used to account for their biological heterogeneity.

To produce ASC-derived conditioned medium (CM), passage 3–6 ASCs were cultured in a T-175 flask until 80–90% confluence was reached. The cells were washed twice with PBS and 15 mL of either ciPTEC serum-free medium (SFM) or X-Vivo medium was added depending on whether experiments with ciPTECs or macropahges were performed respectively. The CM was harvested after 24 h, centrifuged at 2000× *g* 10 min 4 °C, and filtered through a 0.22 µm filter to remove any cell debris. Representative images of ASCs after CM production are shown in Appendix A. The processed CM were then stored in −80 °C for later use. As a control medium, ciPTEC SFM and X-Vivo were also incubated cell-free in empty culture flasks at 37 °C, 5% CO_2_, for 24 h and subjected to the same harvesting process described above.

### 2.2. Ultra-Filtration of CM to Deplete Extracellular Vesicles

To investigate whether soluble factors or rather EVs mediate the secretome effect, we decided to first deplete EVs. To deplete EVs, the CM was filtered through 100 kDa Amicon Ultra-15 centrifugal filter devices (Merck Milipore, Burlington, MA, USA, #UFC810024) that were pre-treated with bovine serum albumin (BSA) to reduce the loss of CM soluble factors in the membrane, as described previously [23]. First, the 100 kDA centrifugal filter devices were washed with EV-grade water as per the manufacturer’s manual. A total of 12 mL of 1% (*w*/*v*) BSA in PBS was then added into the upper compartment of the device and incubated for 2.5 h at room temperature. Subsequently, the upper compartment was washed four times with 15 mL PBS without centrifugation and 1× with centrifugation in swing rotor at 4000× *g*, 10 min, 4 °C. After the washing step, 15 mL of CM was added into the upper compartment of the device and centrifuged until all of the CM was filtered (dead stop). The CM flow-through (CM-FT) in the lower compartment was then collected and subjected to nanoparticle tracking analysis (NTA) to confirm EV depletion. A total of 1 μL of concentrated CM-FT was diluted in sterile-filtered PBS at a dilution of 1:1000 and analyzed using NTA (ZetaView, Particle Metrix GmbH, Inning am Ammersee, Germany; sensitivity 80%, shutter 100, 11 positions, 2 cycles; in collaboration with the Urology department, University Medical Centre, Mannheim).

### 2.3. Thiol Measurement of CM

Free thiols are sulfhydryl groups (R-SH) that can be found in peptides and proteins and are very sensitive to ROS. Therefore, free thiols not only serve as a redox switch in the cells but also act as an ROS scavenger [24]. Given their roles in the cellular redox state, we measured the concentration of free thiols in the CM to predict its anti-oxidative capacity (Thiol Quantification Assay kit, Abcam, Cambridge, UK, # ab112158). The standard and the thiol green indicator solution were prepared according to the manufacturer’s manual. A total of 50 µL of standard solution or CM were incubated with 50 µL of thiol green indicator solution for 10 min in the dark. The fluorescence was read using a microplate reader (TECAN, M200) at ex/em: 490/520 nm.

### 2.4. ciPTEC Culture and Cisplatin Treatment

Conditionally immortalized PTECs (ciPTECs 14.4, stably transfected by SV40T and hTERT, Cell4Pharma) were cultured as previously described [25]. ciPTECs were expanded in DMEM HAM’s F12 (ThermoFisher, Waltham, MA, USA, #11039047), supplemented with 5 μg/mL insulin, 5 μg/mL transferrin, 5 μg/mL selenium (Sigma Aldrich, St. Louis, MO, USA, #11074547001 50MG), 35 ng/mL hydrocortisone (Sigma Aldrich, #H0135-1MG), 10 ng/mL epidermal growth factor (Sigma Aldrich, #E9644-5MG), 40 pg/mL tri-iodothyronine (Sigma Aldrich, #T5516-1MG), and 10% (*v*/*v*) fetal bovine serum (FBS, ThermoFisher, #10270106) in 33 °C, 5% CO_2_, incubator. For the experiments, ciPTECs were seeded at 48,400 cells/cm^2^ (unless indicated otherwise) and kept at 33 °C in a 5% CO_2_ incubator for 1 day. The next day, ciPTECs were cultured at 37 °C, 5% CO_2_, for 7 more days to allow for cell maturation.

Once matured, ciPTECs were treated with 15 µM cisplatin (Cis, Cisplatin Teva^®^, Parsippany-Troy Hills, NJ, USA) in ciPTEC SFM for 1 h. After 1 h, the medium was replaced with CM containing 15 µM cisplatin and the cells were incubated for another 23 h. ciPTECs treated with control medium (ciPTEC SFM) served as the untreated (UT) and cisplatin-treated control (CTRL), respectively.

### 2.5. ciPTEC Viability

After a total of 24 h of treatment with cisplatin, ciPTEC viability was measured using PrestoBlue™ HS (Invitrogen, Waltham, MA, USA, #P50200) as per the manufacturer’s instruction. ciPTECs were washed once and HBSS containing 10% (*v*/*v*) of PrestoBlue was added. ciPTECs were then incubated for 30 min at 37 °C, before being measured using a microplate reader (TECAN, Männedorf, Switzerland, M200) with ex/em: 560/590 nm. The results are presented as the relative value of the untreated healthy ciPTECs.

### 2.6. ciPTEC Metabolic Activity

To assess the metabolic activity of ciPTECs, the intracellular ATP concentration was measured (CellTiter-Glo^®^ Luminiscent Cell Viability Assay, Promega, Madison, WI, USA, #G7571) as per the manufacturer’s instruction. After 24 h of cisplatin treatment, the reagent was added with the same volume as the media in each well. To induce cell lysis, the cell culture plate was placed on an orbital shaker for 2 min and subsequently incubated at room temperature to stabilize the luminescent signal. The signal was then read using a microplate reader (TECAN, M200). The results are presented as the fold change to the healthy untreated ciPTECs.

### 2.7. ciPTEC Migratory Capacity

To assess ciPTEC migratory capacity, ciPTECs were seeded in ImageLock 96 well plates (Essen BioScience, Sartorius, Göttingen, Germany, #4379) and were matured and treated with cisplatin and the CM for 24 h as explained above. Then, the media were discarded and replaced with HBSS. Scratch wounds were created using a wound maker tool (Essen BioScience, Sartorius, #4563) as per the manufacturer’s instruction, before immediate media change to the CM or ciPTEC SFM, both without cisplatin. The closing of the scratch wounds was then recorded for the next 3 days with 3 h intervals using live cell imaging (Incucyte Zoom, Essen BioScience, Sartorius). The relative wound density (%) was calculated using the respective Incucyte algorithm.

### 2.8. ciPTEC Apoptosis

Previous studies indicated that cisplatin induces either apoptosis or necrosis/necroptosis of cultured PTECs, depending on the dose: continuous low-dose cisplatin exposure (3–50 µM) induces apoptosis, whereas transient high-dose exposure causes necrosis/necroptosis (300–1000 µM) [26]. The findings were reproduced under the chosen experimental setup: apoptosis rather than necrosis/necroptosis occurred in cisplatin-treated ciPTECs [27]. To assess cisplatin-induced apoptosis, live cell imaging was used. CiPTECs were seeded at 5000 cell/cm^2^ 8 days prior to the experiment and underwent subsequent maturation as well as cisplatin and CM treatment, as described previously. After a total of 24 h of treatment with or without CM and Cis, the medium was discarded and cells were washed once with HBSS. Fresh CM (without cisplatin) containing 50 nM Apotracker (Biolegend, #427401) was added, and ciPTEC apoptosis was monitored using live cell imaging (Incucyte SX5, Sartorius) for 3 days with 3 h intervals. Apoptosis was assessed by quantifying the total integrated intensity of green fluorescence (GCU x µm^2^/image), indicating apoptotic cells, normalized by cell confluence (%, percentage of total phase object area) and presented as the relative value to UT-CTRL.

### 2.9. Nephrotoxicity PCR Array

To assess nephrotoxicity, ciPTEC mRNA was harvested 24 h after cisplatin treatment with or without CM and isolated using the miRNeasy Kit (Qiagen, Venlo, The Netherlands, #217084) according to manufacturer’s instruction. The concentration and quality of isolated mRNA were assessed using Tecan Infinite^®^ 200. mRNA was subjected to cDNA synthesis using RT^2^ First Strand Kit (Qiagen, #330401). The cDNA was then analyzed using RT^2^ Profiler PCR Array for nephrotoxicity (Qiagen, #330231) and RT^2^ SYBR Green Mastermix (Qiagen, #330502) according to the manufacturer’s instruction.

### 2.10. Confirmatory RT-qPCR

Validation of PCR array data was performed on independent samples using quantitative reverse transcription polymerase chain reaction (RT-qPCR). Twenty-four hours after treatment, ciPTEC mRNA was isolated using RNeasy Micro Kit (Qiagen, #74004) and the cDNA was generated using SensiFAST™ cDNA Synthesis Kit (Bioline, Memphis, TN, USA, #BIO-65054). Gene expressions of growth arrest and DNA damage-inducible alpha (GADD45a), cyclin-dependent kinase inhibitor 1A (CDKN1a), activating transcription factor-3 (ATF-3), and heme-oxygenase-1 (HMOX-1) were assessed using SensiFAST™ SYBR^®^ No-ROX Kit (Bioline, #BIO-98005). The primers used in this study were obtained from Eurofins (Hamburg, Germany) and the sequences can be found in Appendix A. Glyceraldehyde 3-phosphate dehydrogenase (GAPDH) served as the reference gene. The relative value of the target genes was analyzed using 2^−ddCt^ method and normalized against the reference gene. The results are presented as the relative value to the UT-CTRL.

### 2.11. Intracellular Reactive Oxygen Species Levels

Intracellular reactive oxygen species (ROS) of ciPTECs were measured using 2′,7′-dichlorodihydrofluorescein diacetate (H_2_DCFDA, Invitrogen, #D399). ciPTECs were first loaded with 10 µM H_2_DCFDA in HBSS for 10 min, washed twice with HBSS and incubated with their normal growth media for 30 min at 37 °C, 5% CO_2,_ for recovery. Once the cells recovered, they were treated with cisplatin and the CM (1 h Cis alone followed by 4 h of Cis + CM treatment). ciPTECs treated with 1 µM H_2_O_2_ in ciPTEC SFM served as positive control. Five hours after the treatment, H_2_DCFDA fluorescence representing the intracellular ROS level was measured using a microplate reader (TECAN, M200; ex/em: 495/520 nm). The ROS level is presented as the relative value to UT-CTRL.

### 2.12. Monocyte Isolation and Macrophage Culture

Primary human monocytes were isolated from buffy coats provided by the German Red Cross Blood Donor Service, Mannheim, from healthy blood donors. The Mannheim Ethics Committee II, Medical Faculty Mannheim, Heidelberg University, waived the need for ethical approval; all donors signed informed consent that blood can be used for research and development purposes. Buffy coats were first diluted three times in PBS 2 mM EDTA, then overlaid gently onto a ficoll gradient (Sigma Aldrich, #GE17-1440-02), and centrifuged at 420× *g* for 30 min without a break. Peripheral blood mononuclear cells (PBMCs) were collected from the interphase. Monocytes were then isolated using magnetic-activated cell sorting with anti-CD14 microbeads (Miltenyi Biotech, Teterow, Germany, #130-050-201) as per the manufacturer’s instruction. Isolated monocytes were then cultured at a seeding density of 10^6^ cells/mL in serum-free X-Vivo 10 (Lonza, Basel, Switzerland, #BE04-380Q) supplemented with 10 ng/mL recombinant human macrophage colony-stimulating factor (M-CSF, Peprotech, Cranbury, NJ, USA, #300-25) for 6 days without media change.

After 6 days, macrophages were treated with 15 µM cisplatin in X-Vivo 10 and the CM as described for ciPTECs. Macrophages cultured with X-Vivo 10 in the absence and presence of cisplatin served as untreated (UT-CTRL) and cisplatin controls (Cis-CTRL). Of note, both adherent and non-adherent macrophages were included in the experiment.

### 2.13. Macrophage Surface Marker Expression

Upon cisplatin and CM treatment, macrophages were harvested using accutase (Sigma Aldrich, #A6964) and subsequent gentle scrapping of the culture wells. The harvested macrophages were washed once with PBS 2 mM EDTA and stained for 20 min with the following human antibodies: anti-HLA-DR-PE-Cy7 (Clone: L243, BioLegend, San Diego, CA, USA, #307616), anti-CD86-BV421 (Clone: IT2.2, BioLegend, #305426), anti-CD206-PE (Clone:19.2, BD Biosciences, Franklin Lakes, NJ, USA, #555954), and anti-CD163-BV510 (Clone: GHI/61, BioLegend, #333628). All antibodies were properly titrated before. Consecutively, the cells were also stained with the viability dye Sytox Bue (Invitrogen, #S34857). The cells were then acquired using flow cytometry (FACS Canto II, BD Biosciences) and the obtained .fcs data were analyzed with FlowJo 10 software (purity check, Appendix A; changes in macrophage marker expression, Appendix A).

### 2.14. Macrophage Phagocytosis

The phagocytosis capacity of macrophages was measured by adding 5 µg pHrodo^®^ green *E. coli* Bioparticles^®^ (Essen BioScience, #4616) per 10^5^ macrophages. Subsequently, phagocytosis was measured by live cell imaging for 6 h with 15 min intervals. Phagocytosis was determined by calculating the total integrated intensity of green fluorescence (GCU × µm^2^/image and normalized by cell confluence (%)). The data are shown as a relative value to UT-CTRL.

### 2.15. Indirect Co-Cultures of ciPTEC and Macrophages

Before seeding ciPTECs on transwell membranes, transwell inserts (0.4 µm transparent PET Membrane (Falcon, New York, NY, USA, #353095)) were coated with 2 mg/mL L-DOPA (Sigma Aldrich, #D9628-5G) for 4 h in 37 °C and subsequently with 25 µg/mL of human collagen IV (Sigma Aldrich, #C6745-1ML) for 1 h in 37 °C [25]. After coating, the membranes were washed three times with HBSS. A total of 4 × 10^4^ ciPTECs/transwell were then seeded onto the apical side of the membrane. A total of 300 µL and 700 µL of ciPTEC growth medium were added to the apical and basolateral side, respectively. The seeded ciPTECs were left in the transwell membrane for 3 days at 33 °C, before being moved to 37 °C for maturation for an additional 7 days. The ciPTEC medium was refreshed periodically.

During the maturation of ciPTECs, monocytes were isolated and seeded in a 24-well plate (Falcon, #353504) at a density of 2 × 10^5^ cells/well and matured in X-Vivo-10 supplemented with 10 ng/mL M-CSF at 37 °C for 6 days without media change.

After the maturation, ciPTECs and macrophages were treated separately with 30 µM cisplatin in ciPTEC SFM and X-Vivo-10 + M-CSF, respectively, for 1 h. The cisplatin concentration had to be doubled in the transwell system to achieve a similar level of apoptosis, as apparently the differing cell culture/insert plastic led to a lower cisplatin toxicity. After 1 h of initial treatment with cisplatin, ciPTEC transwells were transferred to the well plates containing the macrophages (cell ratio 1:5). The co-cultures were then treated with either the CM (pooled from 3 different ASC batches/donors, produced in ciPTEC SFM and X-Vivo-10 + M-CSF, for the apical and basolateral side, respectively) or the respective control medium with or without 30 µM cisplatin for another 23 h. As a control, ciPTECs and macrophages were cultured alone and treated in the same manner as the co-culture.

After a total of 24 h of cisplatin treatment, the supernatant (SN) from the apical and basolateral side was collected, spun down to remove cell debris, and cryopreserved at −80 °C. The ciPTECs on the transwell membrane were subjected to staining (see below), while the macrophages were subjected to phagocytosis assay (see above).

### 2.16. Staining of ciPTECs on Transwell Membrane

Because culture of the transwell did not allow for apoptosis measurement using live cell imaging, immunofluorescence was used. After ciPTEC inserts were removed from the co-culture and washed once with HBSS, the cells were incubated with 150 nM Apotracker in HBBS at 37 °C for 20 min. Then, excess Apotracker was washed off twice with HBSS, and the cells were fixed with 4% paraformaldehyde (PFA) for 10 min. Once fixed, the cells were washed once with PBS and the membrane was carefully cut from the insert using a 24 G needle. The membrane was then washed again twice with PBS, and blocked with blocking solution (10% FBS + 0.15% Triton-X in PBS) for 1 h. The membrane was stained with rabbit anti-human connexin-43 (CX43; Sigma, Ronkonkoma, NY, USA, #C6219, 1:1000) in blocking solution overnight. The next day, the membrane was washed three times with PBS with 5 min of incubation per wash, before incubating with the goat anti-rabbit secondary Ab AF568 (Life Technologies, Carlsbad, CA, USA, #A-11011) for 30 min in the dark. After washing three times with PBS, the membrane was stained with 300 nM DAPI (SantaCruz, Dallas, TX, USA, #SC-300415) in PBS for 5 min and washed again three times before being mounted on a microscope slide.

The image of the ciPTECs was acquired using Zeiss LSM 800. ImageJ Fiji was used to quantify the fluorescence intensity of CX-43, the number of Apotracker spots (using Plugin TrackMate7 [28]), and the number of nuclei fragments, fragmented and intact nuclei. The resulting data were normalized against the number of intact nuclei and presented as the relative value of the untreated ciPTECs cultured alone.

### 2.17. Multiplex Assay to Quantify Secreted Factors in the Co-Culture System (Basolateral Side)

To assess the secreted factors derived from the macrophages within co-cultures harvested from the basolateral (macrophage) side of the co-culture, a 22-plex panel assay (ProcartaPlex Thermo Fisher Scientific, #PPX-22) was used to assess the following: Arginase-1, Fractalkine (CX3CL1), Granulocyte-macrophage colony-stimulating factor (GM-CSF), Hepatocyte growth factor (HGF), Heat shock protein 60 (HSP60), Interferon-β (IFN-β), Interleukin-1β (IL-1β), IL-1RA, IL-8 (CXCL8), IL-10, IL-13, IL-33, IFN-γ, inducible protein-10 (IP-10), M-CSF, Matrix metallopeptidase 9 (MMP-9), Platelet-derived growth factor-BB (PDGF-BB), Chemokine (C-C motif) ligand 5 (CCL5), S100A8/A9, Survivin (BIRC5), Tissue inhibitor of metalloproteinase-1 (TIMP-1), TNF-α, and Vascular cell adhesion molecule 1 (VCAM-1). The kit was first equilibrated to room temperature. The dilution of the standard solution and the loading of the beads and supernatant were conducted as per the manufacturer’s manual. The intensity of each marker in the samples was then acquired using Luminex 200 Multiplex Bead Array (Merck-Millipore) and Luminex xPonent Version 3.1 software. The standard calibration was performed with the Logistics 5P Weighted method.

### 2.18. TNF-α Measurement in the Co-Culture System (Apical Side)

The concentration of TNF-α of the co-culture supernatant on the apical (ciPTEC) side was measured using human TNF-α DuoSet ELISA (R&D Systems, Minneapolis, MN, USA, #DY210) as per the manufacturer’s instruction.

### 2.19. Statistical Analysis

The statistical analysis was performed using GraphPad Prism 9.2.0 software (GraphPad Software). Data are presented as mean ± standard deviation. *N* refers to the biological replicates for each donor per cell type, and *n* indicates the independent technical replicates. The statistical analyses run for each data set and the number of replicates are indicated in the figure legends. Statistical significance is indicated as the following: * *p* < 0.05, ** *p* < 0.01, *** *p* < 0.001, and **** *p* < 0.0001.

## 3. Results

In this study, we aimed to investigate the renoprotective capacity of the ASC secretome in the context of renal cell and macrophage interaction (Figure 1A). First, the effect of the ASC secretome on ciPTECs and macrophages was studied separately to establish the ASC therapeutic effect on each cell directly and to gain insights into the mechanisms of action. Then, the ASC secretome capacity in modulating the interplay between ciPTECs and macrophages in transwell co-cultures of both cell types was assessed.

### 3.1. Cisplatin Decreases ciPTEC Viability, Metabolic Activity, and Migratory Capacity

Hypothesizing that the ASC secretome will protect rather than cure from cisplatin toxicity, a setting was chosen where ciPTECs were treated with cisplatin for 1 h before adding ASC-CM (Figure 1B). To calculate the concentration of cisplatin with only moderate cytotoxicity but high functional impact that can be rescued by the CM, a titration of cisplatin was performed, and viability, metabolic activity, and migratory capacity in a wound scratch assay were assessed. After 24 h of treatment, all features were dose-dependently reduced (Figure 1C–E). A concentration of 15 µM cisplatin was chosen because it impaired viability and metabolic activity only by 20–25% while exerting a highly pronounced effect on migratory capacity (76 ± 14% viability, 80 ± 19% metabolic activity, and 36 ± 20% migratory capacity compared to the untreated control).

#### 3.1.1. ASC Secretome Attenuated Cisplatin-Induced Cell Toxicity and Promoted Migration of ciPTECs by Ameliorating Apoptosis, DNA Damage, and Oxidative Stress

We first sought to investigate the effect of ASC secretome on cisplatin-treated ciPTECs (Figure 2A). Twenty-four hours after treatment, cisplatin decreased ciPTEC viability by 13%; meanwhile, the CM significantly increased ciPTEC viability in cisplatin-untreated (UT-CM) and treated (Cis-CM) groups (Figure 2B). The protective effect of the ASC secretome became even clearer when assessing apoptosis of cPTECs challenged with cisplatin for 24 h and cultured for a further 3 days in CM or in CTRL medium. While cisplatin induced significant apoptosis, ASC CM reduced this significantly by 55.68% (Cis-CM, Figure 2C,D, Appendix AA). Not only preventing cell death, the CM also promoted ciPTEC migratory capacity (Figure 2E,F, Appendix AB). This effect was prominent in both the untreated and treated CM condition, while cisplatin per se affected wound closure only slightly.

To understand better the molecular pathways by which cisplatin and the CM exert their effects, a nephrotoxicity PCR array was performed. Cisplatin affected ciPTECs’ gene expression profile, indicating nephrotoxicity (Figure 3A). Meanwhile, the CM ameliorated the cisplatin effects largely. Genes affected by cisplatin and modulated by the CM included mainly genes involved in apoptosis and oxidative stress (HMOX1, NQ01, GATM, GPX, BTG2, TNFRSF12A, GADD45A, BMP1, and SOCS3). Further, cell cycle/cell proliferation-associated genes (UCHL1, ATF3, GADD45A, HSP90AA1, ODC1, BMP4, CXCl1, and TIMP1), genes involved in xenobiotic metabolism (GSTP1, SCD, and RGN), and genes involved in tissue remodeling/extracellular matrix/cytoskeleton regulators (CCDN1, FGB, CST3, TIMP1, CD24, VCAM1, FN1, G6PC, and CD44) were affected. Independent PCR validation documented that CDKN1a and GADD45a involved in cell cycle control and DNA damage, HMOX-1 regulating oxidative stress, and ATF-3 indicating cellular stress were all elevated by cisplatin treatment but ameliorated upon ASC CM treatment (Figure 3B–E). Interestingly, genes modulated by CM treatment already within the CTRL setting involved genes in metal ion binding, transporters, and xenobiotic metabolism (FMO2, CTSS, SLC22A5, HMOX2, CCS, HMOX1, RGN, BHMT, CP, CYP2D6, GSTK1, SLC22A6, CYP2C19, IDH1, and MT1A). Further, the CM modulated regulators of oxidative stress (HMOX1, NQO1, GPX2, and SOD3), of apoptosis (RTN4, ANGPTL4, SOD2, and CD44), and of cell cycle/cell proliferation (CCNG1, GPNMB, ANGPTL4, MCM6, BMP4, CXCL1, TIMP1, EGF, and CD24). This suggests that factors within the conditioned medium modulate cellular functions largely and further ameliorate cisplatin nephrotoxicity.

#### 3.1.2. Anti-Apoptotic Effect of CM Was Not Affected by EV Depletion but Related to Its Free Thiol Content

As previous reports suggested EVs to mediate the MSC renoprotective effects in an animal model of AKI [29,30], we postulated that EV depletion would neutralize the protective effect of the CM. CM from different ASC batches/donors contained a wide variety of particle concentrations, effectively depleted by ultra-filtration as seen in CM-flow through (CM-FT) (Figure 4A). Cell-free CTRL medium before and after ultra-filtration contained no measurable particles. Yet, EV depletion did not neutralize the protective effect of the CM on inhibiting apoptosis (Figure 4B), indicating that EVs were not involved in the protective activity of the CM. Of note, to compensate for the eventual depletion of soluble factors by ultra-filtration as suggested by Whittaker et al. [23], the apoptotic level of cis-CM and cis-CM-FT were normalized to their respective CTRL.

Having shown that the protective effect of the CM was not mediated by EVs, we hypothesized that free thiols acting as ROS scavengers [24,31] may have ameliorated oxidative stress, as indicated by the gene expression data that showed a variety of genes involved in oxidative stress modulated by the CM (Figure 3B). Indeed, the CM had a higher free thiol content as compared to the empty media, yet showed large donor-dependent differences in concentration (Figure 4C). Supporting our hypothesis, the CM significantly decreased ROS level in both UT- and Cis-treated ciPTECs by 43% and 79% in UT-CM and Cis-CM, compared to UT-CTRL and Cis-CTRL, respectively (Figure 4D). Based on a Spearman analysis, free thiols content in the CM inversely correlated with ciPTEC apoptosis (R = −0.8, *p* = 0.167), although not significantly due to the large donor heterogeneity and too low number of biological replicates for this kind of analysis (Appendix A).

### 3.2. MSC-CM Promoted Macrophage Polarization towards the M2-like Phenotype

Having documented the protective effect of the CM on cisplatin injury in ciPTECs, we next aimed to assess its effect on macrophages (Figure 5A). First, phenotype skewing and phagocytosis was assessed. Unlike in ciPTECs, cisplatin per se barely affected macrophages (Figure 5B–G, Appendix A). The M1 surface marker expression (HLA-DR, CD38, and CD86) of macrophages remained similar, as did the M2 marker expression (CD206, CD163, and phagocytosis). The CM, however, exerted significant effects, yet irrespective of cisplatin. The CM significantly reduced the expression of the M1 markers HLA-DR and CD86 and increased the M2 markers CD206 and CD163. Only In the presence of the CM did cisplatin significantly attenuate the CM-induced CD206 expression. The CM significantly increased macrophage apoptosis, which was slightly reduced upon cisplatin treatment.

### 3.3. Macrophages Did Not Boost the Protective Effect of MSC-CM on ciPTECs

Our data show (1) that cisplatin exerted nephrotoxic effects on ciPTECs, which were significantly ameliorated by the CM, and (2) that while cisplatin had no effect on macrophage polarization and function, the CM was able to induce a shift to the more anti-inflammatory M2 state. Based on this anti-apoptotic and anti-inflammatory action, we postulated that the CM could elicit an even more pronounced effect in co-cultures of ciPTECs and macrophages by modulating the vicious crosstalk of injured ciPTECs acting on macrophages and macrophages acting as anti-inflammatory agents on ciPTECs. Accordingly, indirect co-cultures of ciPTECs and macrophages were established (Figure 6A). Of note, in the transwell system, the concentration of cisplatin needed to be doubled to 30 µM, as 15 µM of cisplatin did not cause the same level of apoptosis as in the ciPTEC monoculture (Appendix A). As ciPTECs were now seeded on inserts, live cell imaging of apoptosis was not possible, and thus immunofluorescence was performed. Next to Apotracker staining and nuclear fragmentation, which are indicative for apoptosis, connexin-43 (CX-43) was analyzed, a gap junction protein implicated in various nephropathologies [32]. Cisplatin treatment increased CX-43 expression levels 4 ± 3.46- and 3.2 ± 1.32-fold in ciPTECs monoculture and co-culture with macrophages, respectively (Figure 6B,C, compared to UT-CTRL). Co-cultures showed a small decrease in CX-43 levels, but the CM, which reduced CX-43 levels 1.9 ± 0.65- and 1.97 ± 0.83-fold in ciPTEC mono- and co-cultures, respectively, overruled this reduction. Despite the use of CM pooled from each three donors in these experiments, large data heterogeneity was observed in the transwell system, resulting in non-significant statistical analyses. Variances were apparently lowered upon macrophage co-culture. Possibly by chance, highly potent ASC-CM donors were pooled in one experiment, while in the other experiment, donor pooling leveled out potent and less potent donor attributes. Despite the lower variation in data, the co-culture had no apparent effect on CX-43 expression. While the co-culture per se lowered cisplatin-induced apoptosis, the anti-apoptotic properties of the CM completely surpassed this co-culture effect (apoptotic levels were 2.42 ± 1.08- and 2.44 ± 1.49-fold for cis-CM in ciPTEC mono- and co-cultures, respectively, Figure 6D). When assessing nuclei fragmentation, the CM also exerted a greater effect in suppressing the number of fragmented nuclei and nuclei fragments than that of the co-culture (Figure 6E).

Furthermore, given the importance of TNF-α in cisplatin-induced AKI, TNF-α levels in the supernatant were measured from the apical ciPTEC side of the co-culture. The increase in TNF-α level caused by cisplatin treatment in the co-culture setting was higher in mono- than co-cultures (12.76 ± 12.34-fold and 10.59 ± 10.59-fold, for Cis-CTRL in ciPTEC mono- vs. co-cultures, respectively, Figure 6F), indicating a regulatory crosstalk between ciPTECs and macrophages in the indirect co-culture. TNF-α level in both culture settings was further suppressed by CM treatment to 5.99 ± 5.02- and 4.76 ± 4.33-fold in mono- and co-cultures, respectively.

#### Despite Enhancing the Phagocytosis Capacity of Macrophages, CM Attenuated Macrophage Factor Secretion Triggered by ciPTECs

We initially postulated that ASC-CM treatment could prevent the vicious crosstalk of injured ciPTEC recruiting and activating pro-inflammatory macrophages (Figure 7A). Yet, the macrophage presence appeared to mediate only minor effects on cisplatin nephrotoxicity in ciPTECs, being highly overruled by the CM. Addressing now the macrophages in the co-cultures revealed an increase in phagocytosis activity by 47% (UT-CTRL, Figure 7B). However, again, the stimulatory effect of the CM entirely overruled this (4.37 ± 3.09- and 4.32 ± 2.59-fold for UT-CM of macrophage mono- and co-cultures, respectively). While 15 µM cisplatin showed no effect on macrophage viability and function (Figure 5G), the increased concentration of 30 µM cisplatin in the co-culture system compromised phagocytosis slightly (0.72 ± 0.24- and 0.9 ± 0.39-fold for Cis-CTRL of macrophage mono- and co-cultures, respectively). Interestingly, this cisplatin inhibition of phagocytosis also persisted in the presence of the CM (2.80 ± 1.13- and 2.62 ± 0.65-fold for Cis-CM of macrophage mono- and co-cultures, respectively).

To address the postulated changed crosstalk between ciPTECs and macrophages in more detail, the supernatant from the basolateral side of the co-culture was probed for the analytes listed in Appendix A. Most factors were clearly macrophage-derived and hardly detectable in ciPTECs (Figure 7C, Appendix AA–T; data are presented as the relative value to the untreated macrophage monoculture). Co-cultured ciPTECs generally increased the secretion of certain macrophage factors, including DAMPs. For instance, a 71%, 61%, 44%, and 30% increase in arginase-1, fractalkine, and IFN-β was observed. CCL5 and IL-1β, respectively, were observed upon cellular crosstalk. Some factors remained unchanged including HSP60, MMP9, IP10, IL-8, IL-1Ra, GM-CSF, and S100A8/9; meanwhile, the rest showed between a 10–20% increase stimulated by the co-culture. Cisplatin, on the other hand, decreased the secretion of all of those factors, significantly for IL-1b, IL-10, IFN-b, IL-13, and arginase. IL-8 and MMP-9 were, however, elevated in macrophage monocultures by 74% and 19%, respectively. This was not observed in the co-cultures. Interestingly, ciPTECs cultured alone were also able to produce IL-8 into the basolateral side; however, in contrast to macrophages, the IL-8 release by ciPTECs was downregulated by cisplatin. Therefore, the lower IL-8 concentration seen in the cisplatin-treated co-culture as compared to the untreated counterpart might be caused by the opposing effect of cisplatin in those two cell types. The other factors that were secreted in the ciPTEC monoculture and modestly elevated by cisplatin include the DAMPs arginase-1 and fractalkine, and IL-33. Yet, IL-33 was detected in both basal media and remained unchanged upon conditioning. We cannot exclude that this was an artefactual detection.

The CM further attenuated factors secreted in the co-cultures. In general, the CM downregulated the majority of secreted factors. This was most prominent for macrophages alone and co-cultures, significantly, for instance, for IL-1β, MMp9, IP-10, IL-10, IFN-β, Fractalkine, CCL5, GM-CSF, TNF-α, arginase, survivin, and S100A8/9. HGF and PDGF-BB that were contained in the CM itself did not show such regulation. Specifically, the CM produced in X-Vivo medium (for macrophages) contained high HGF and PDGF-BB levels (Figure 7C), higher than those in the CM produced in ciPTEC SFM. IP10 was significantly reduced in the co-cultures when treated with the CM.

Lastly, long-term cisplatin treatment reduced macrophage viability, as shown in Appendix AA, where cisplatin-treated macrophages showed more cell debris than their untreated counterparts did. The constant presence of cisplatin in both the Cis-CTRL and Cis-CM groups reduced macrophage confluence, which started even from day 2 (Appendix AB,C), indicating that despite the CM effect in enhancing phagocytosis, the CM could not attenuate macrophage death induced by cisplatin. Given the strong association of free thiol content in the anti-apoptotic capacity of the CM, the free thiol content in the CM used to treat macrophages was measured (Appendix AB). Interestingly, while CM for macrophages contained a higher concentration of cytokines and growth factors than that for ciPTECs (Figure 7C), its free thiol concentration was 4.5 times lower than the CM for ciPTECs (1.9 ± 2.12 µM vs. 8.59 ± 8.8 µM, for the CM intended for macrophages and ciPTECs (Figure 4C), respectively). Free thiols in the X-Vivo CM showed no trend to correlate to its phagocytosis-inducing capacity (Appendix AB).

## 4. Discussion

In this study, we have demonstrated the protective effect of the MSC secretome (ASC CM) against cisplatin-induced injury in both ciPTECs and macrophages, first individually and second within co-cultures (Figure 8). Yet, contrary to our hypothesis, we did not observe the added benefit of the CM in co-cultures, counteracting the vicious cycle of cisplatin-induced nephrotoxicity by modifying the crosstalk of PTECs and macrophages. Yet, this may not rule out the possibility that MSCs and their secretome beneficially influence this complex interplay in vivo.

The protective effect of ASC CM on ciPTECs involved the suppression of apoptosis and the enhancement of migratory capacity. The CM partially reversed the expression of nephrotoxicity genes induced by cisplatin. Examples are CDKNA1 and GADDH45a, both induced by various cellular stresses, such as DNA damage and oxidative stress, contributing to cell cycle arrest [33,34,35]; HMOX1, the gene encoding for heme oxygenase-1, an important anti-oxidative defense mechanism [36]; and ATF-3, triggered by ER stress, chemokines, and cytokines regulating apoptosis and necrosis [37]. The attenuation of cisplatin-induced upregulation of these genes documented that the CM rescued ciPTECs by alleviating cellular stress, especially oxidative stress.

Given the pivotal role of oxidative stress in ciPTEC apoptosis, the anti-oxidative capacity of the CM was investigated. Free thiols or sulfhydryl groups (R-SH) are very easily reduced by ROS and therefore reflect the fluctuation of the redox state of their environment [24]. Due to their sensitivity to ROS, free thiols not only serve as a good indicator of the progression of various degenerative diseases (e.g., digestive, respiratory, cardiovascular, metabolic, and cancer diseases) but also play an important role in dampening oxidative stress by scavenging ROS. The treatment of ciPTECs with the CM led to a remarkable decrease in intracellular ROS levels and subsequently reduced apoptosis. In fact, our data suggest an inverse relationship between free thiol content and the degree of apoptosis suppressed by the CM. It is noteworthy that a high variance in free thiol content was present across MSC donors. We attribute the ability of MSCs to produce a high level of free thiols to the cell fitness, their vitality, their doubling time, cell size, and morphology; however, this hypothesis should be tested in a larger study with a higher number of MSC donors. Given the importance of free thiols in cellular redox homeostasis, measuring free thiols presents an easy, robust, and reliable option of a potency assay to select or predict the anti-oxidative properties of MSCs.

Within an accompanying study, we investigated in more detail the underlying mechanisms [27]. Various pathways have been described by which either cisplatin causes renal injury or by which the MSC/MSC secretome can rescue it from nephrotoxic injury. We tested various inhibitors for pathways that have been shown to be involved in cisplatin-induced apoptosis, necrosis, or necroptosis [30,38]. N-acetylcysteine (NAC), as expected from the data presented herein, reduced the number of apoptotic cells in both controls, but significantly more in cisplatin-injured ciPTECs. Yet, there was no apparent inhibition of ASC CM effects, suggesting that its anti-oxidative properties exceeded NAC’s inhibitory action. Necrostatin (NEC) is an inhibitor targeting receptor-interacting serine/threonine-protein kinase 1 (RIPK1, aka RIP1), and the necroptotic pathway. Necrostatin attenuated cisplatin-induced apoptosis, yet again not lowering ASC CM’s anti-apoptotic effects. Likewise, the sphingosine kinase inhibitor (SKI-II) protected against cisplatin-induced apoptosis, while interestingly, Mithramyin A, an inhibitor of Specificity Protein 1 (Sp1) and main regulator of sphingosine kinase 1, had no effect at all. The inhibition of stanniocalcin by a blocking antibody, as suggested by the group of Prockop et al. [38], had no significant effect at all, supporting their notion that this pathway becomes activated merely upon the priming of MSCs. Yet, we observed that the downregulation of miR-181-5p expression appeared to be involved in the anti-apoptotic activity of ASC-CM (Scaccia et al.; manuscript in preparation).

Within the composition of the CM, EVs have been reported as the mediator by which MSCs exert their therapeutic effect [39]. However, we showed that EVs were not the key mediator for the anti-apoptotic property of the CM, suggesting that soluble factors or smaller EVs (<100 kDa) are responsible for this effect. Our finding is in agreement with a recent study by our group showing that EVs are dispensable, e.g., for angiogenesis of endothelial cells [40], and that the bioactivity of EVs is often a result of EV impurity [23,41].

When assessing macrophages alone, previous findings were reproduced showing that MSC CM can promote M2 polarization of macrophages, as seen by the decreased expression of M1 surface markers such as HLA-DR and CD86; the increased expression of M2 markers, including CD163 and CD206; and higher phagocytosis capacity (Figure 5). In a previous study, we showed IL-6 [42], prostaglandin-2 (PGE-2), and transforming growth factor beta 1 (TGF-β1) to be partly but not entirely involved [27]. On the other hand, cisplatin treatment compromised the M2-promoting effect of the CM, suggesting cisplatin uptake by macrophages. Supporting this, alveolar macrophages express the transporters capable of uptaking cisplatin, such as MATE, OCTN1, and OCTN2 [43]. Also, THP1, a human monocyte cell line, polarized into M2 macrophages showed significantly higher cisplatin uptake, as compared to the M1 and M0 counterparts and the monocytic cells per se [44]. This might explain why cisplatin partly compromised the effect of the CM in skewing macrophage polarization towards M2 macrophages. Considering that M2 macrophages are fueled by oxidative phosphorylation in mitochondria [45,46] and that MSCs promote this metabolic shift [47], cisplatin’s ability to affect mitochondrial function [48] could eventually hinder the M2 polarization promoted by the CM.

Beyond the ability to improve the phagocytosis of macrophages, CM could not ameliorate cisplatin-induced macrophage death, at least in the long-term. The measurement of free thiols revealed that the CM produced using X-Vivo 10 for macrophage culture contained 4.5 times lower free thiols than the CM produced in ciPTEC SFM, and free thiol levels were not as apparently related to the increased phagocytosis activity as with apoptosis amelioration (Appendix A). Whether or not the lower free thiol level of the CM in X-Vivo is the reason and why it cannot attenuate macrophage death by cisplatin is beyond the scope of this study.

Contrary to our hypothesis, the co-culture system failed to reveal the expected strong additive effect of the CM on co-cultures, interrupting the vicious cycle of injured ciPTECs and macrophages (Figure 8). Despite the use of pooled ASC CM to reduce donor heterogeneity, the in vitro data were not as robust in the transwell system as in the standard culture system, yet were slightly homogenized in the presence of macrophages. In the absence of the CM, the ciPTEC and macrophage co-culture showed a slight trend of injury attenuation, which was shown by slightly decreased ciPTEC death markers and TNF-α secretion in the co-culture groups compared to the ciPTEC monoculture. However, the protective effect of CM surpassed this co-culture benefit. This might be caused by the fact that CM’s protective effect on ciPTECs was already very strong, eventually overruling the macrophages’ regulatory effect. The CM not only suppressed ciPTEC apoptosis and nuclei fragmentation but also inhibited the expression of CX-43, which has been shown to be an important player in renal fibrosis and CKD progression [49,50]. Furthermore, a reduction in TNF-α secretion by ciPTECs was also observed in the co-culture groups and, even more so, in the CM-treated groups, suggesting that the CM ameliorates also the inflammatory response induced upon cisplatin injury. Of note, the attenuation of TNF-α secretion might in turn contribute to lower cell death as it also plays an important role in promoting apoptosis through the activation of the death receptor or extrinsic pathway [51,52,53].

Nonetheless, given the profound increase in phagocytosis elicited by the CM, the role of macrophages upon CM treatment might be more significant in an in vivo model, as phagocytosis of dead cells is an important step to prevent prolonged inflammation and stimulate regeneration [54,55]. In fact, in a mouse unilateral ureteral obstruction (UUO) model that develops renal fibrosis, compared to the sham control, the population of highly phagocytotic macrophages (M2) decreased, while the ones with low phagocytosis ability increased (M1) [55]. Interestingly, the infusion of highly phagocytic macrophages (M2) attenuated renal fibrosis in this model. It is noteworthy to mention that M1 and M2 are an oversimplification of macrophage polarization showing the extremes; however, macrophage polarization during renal injury is more nuanced [13]. For instance, it has been postulated that the M2 macrophages responsible for renal fibrosis express CD206, CD204, and HLA-DR^high^; meanwhile, the ones responsible for tissue homeostasis express CD206, CD163, and HLA-DR^low^ [56]. Given that our CM boosted macrophage phagocytosis and increased the expression of CD206 and CD163 while decreasing the level of HLA-DR and CD86, CM treatment for cisplatin-induced AKI might be beneficial to prevent the progression of renal fibrosis in vivo. Moreover, the phagocytosis of dead cells can also stimulate macrophages further to promote tubular recovery [55].

Our study demonstrated that in co-cultures, ciPTECs and macrophages could sense and respond to each other’s stimuli. In co-culture, macrophages were activated by ciPTECs, as indicated by the increased production of various factors comprising cytokines (IL-8), chemokines (Fractalkine), DAMPs (arginase-1), growth factors (HGF), lipoxin (survivin), and matrix metalloproteinases (MMP-9) (Figure 7C). This is in line with a previous study revealing that both direct and indirect co-cultures of PTECs and monocytes resulted in the inhibition of monocyte maturation into dendritic cells, marked by lower levels of HLA-DR and CD86 and elevated levels of phagocytosis [57]. Interestingly, previous studies reported injured PTECs to drive macrophage polarization into M1 by releasing EVs containing miR-199a-5p, miRNA-23a, and miR-19b-3p, respectively [58,59], and to switch macrophage metabolism toward glycolysis [60]. In our co-culture system, cisplatin induced pro-inflammatory TNF-α secretion of ciPTECs but slightly decreased it within co-cultures (Figure 6F). TNF-α most likely also affected macrophage polarization and, to some degree, contributed to the downregulation of macrophage phagocytosis in co-cultures treated with cisplatin. The co-culture mediated a decrease in TNF-α secretion which is in contrast to a previous study where macrophages augmented the albumin-induced cytokine release by PTECs [61]. It is also noteworthy that we observed similar patterns regarding TNF-α secretion and cisplatin-induced ciPTEC death markers, all slightly reduced in the co-culture setting, compared to monocultures, implying at least some regulatory effect.

In addition to dramatically increasing phagocytosis, CM downregulated overall macrophages’ factor secretion (Figure 7C). The MSC secretome is known to suppress the secretion of pro-inflammatory cytokines on macrophages stimulated by LPS [62], which is in line with our observation that the CM decreased macrophage pro-inflammatory cytokine and chemokine secretion (e.g., IL-1β, IL-13, IL-8, TNF-α, and IP-10). However, not only decreased pro-inflammatory cytokines, our data showed that the CM also decreased anti-inflammatory cytokines and enzymes such as IL-10, IL-1RA, and arginase-1. This is in contrast to a previous finding that reported that MSC treatment increases the production of IL-10 and arginase-1 [63]. This discrepancy might be caused by the fact that in most of the previous studies, macrophages were also stimulated with LPS [63,64] or M1-polarizing medium [64], which might be a stronger stimulus than co-culture with ciPTECs alone. The upregulation of those anti-inflammatory and pro-regenerative factors in macrophages was observed mostly in in vivo models with higher cell interaction and thus more macrophage stimulation [65].

The strength of our study is that it provides an overview of the interaction between PTECs and macrophages under cisplatin treatment with and without CM. We also shed light on the importance of free thiols in the CM as a metric to assess the anti-oxidative properties of the CM, as well as the profound effect of cisplatin on macrophage viability and phagocytosis that is often overlooked in cisplatin-induced AKI. While there are already many reports on how cisplatin influences inflammatory response in PTECs, only a few have focused on macrophages in the context of kidney injury. The limitation of our study is the mere in vitro design, which can only reflect some aspects of the complex interplay and will not allow us to capture the whole PTEC and macrophage crosstalk comprehensively. Further, within this study, we focused on a CM derived from non-stimulated ASCs. Priming/licensing by either (pro-) inflammatory molecules, hypoxia or other triggers, or cell–cell contacts may further augment therapeutic potency [66].

Our data suggest that MSC CM could ameliorate the cytotoxic effects of cisplatin in AKI, acting predominantly on ciPTECs but also polarizing macrophages to the anti-inflammatory and pro-regenerative M2 phenotype. A therapeutic scenario, therefore, could be to administer MSC CM to patients concomitantly with cisplatin. However, implementing this kind of therapy may intervene with the desired anti-cancer benefit of cisplatin. Yet, there are already ideas on how to concentrate the protective MSC effect locally. To achieve tissue specificity, a matrix metalloproteinase-2 (MMP-2) sensitive self-assembling peptide hydrogel containing MSC EVs has been described. Since MMP-2 is normally upregulated upon kidney injury, the injection of MSC-EV in this hydrogel resulted in local retention in the kidney [67]. Of course, the feasibility of this strategy to limit off-target MSC treatment still needs to be further tested, especially if there are other organs undergoing fibrosis with elevated MMP-2 levels. Nevertheless, this presents an option by which MSC therapy might be applied to prevent or treat cisplatin-induced AKI.

## 5. Conclusions

Our results show that first, ASC CM protected ciPTECs from cisplatin-induced injury, likely associated with its anti-oxidative properties. Second, the CM promoted M2 polarization, which was compromised, however, by cisplatin. Third, our co-culture showed that cisplatin only slightly altered the crosstalk between ciPTECs and macrophages. Contrary to our hypothesis, we did not observe the added benefit of the CM counteracting the vicious cycle of cisplatin-induced nephrotoxicity, at least not in our in vitro setup. This might be caused by the fact that the CM already exerted a very strong protective effect, surpassing the macrophage contribution in this system. Given the short-term in vitro nature, our system may not describe the role of the CM and its effects on the interplay between PTECs and macrophages in induced proximal tubule recovery in its entire complexity. For clinical implementation of the MSC secretome in the future, these aspects should be considered as well.

## Figures and Tables

**Figure 1 cells-13-00121-f001:**
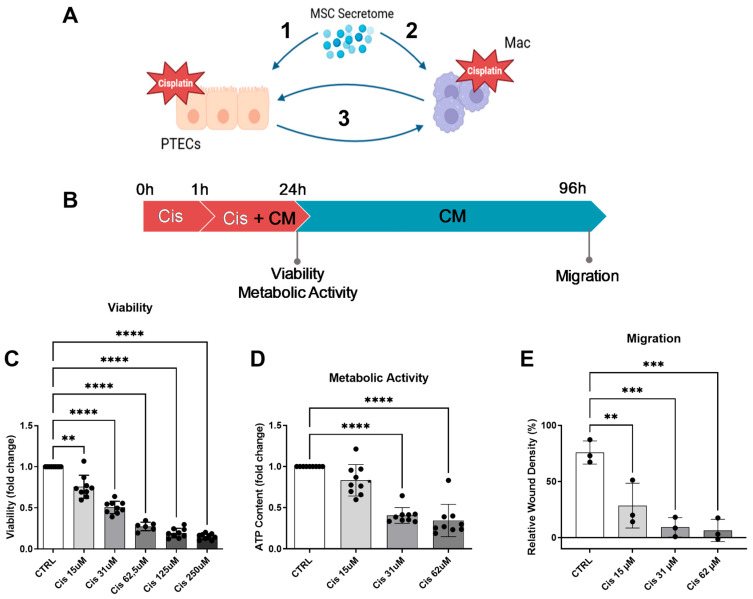
Hypothesis and establishment of the cisplatin injury model. We hypothesize that the MSC secretome acts on various levels to ameliorate cisplatin nephrotoxity: 1—prevention of apoptosis of ciPTECs, 2—prevention of cisplatin toxicity in macrophages and polarization towards M2-macrophages, and 3—boosted effect by preventing the vicious cycle of crosstalk between injured ciPTECs and recruited M1-polarized macrophages (**A**) (image created using BioRender). Matured ciPTECs were treated with different concentrations of cisplatin for 24 h—for experiments with CM, CM was added after 1 h for a total cisplatin treatment time of 24 h (**B**). After 24 h, their viability (**C**) and intracellular ATP were measured (**D**). To assess their migratory capacity, 24 h post cisplatin treatment, the ciPTEC monolayer was scratched and the cisplatin-containing medium was discarded and replaced with fresh SFM. The closing of the wound was monitored using live cell imaging and quantified 3 days after the wound was made (**E**). One-Way ANOVA with Tukey’s multiple comparisons test were used; significance values: ** *p* < 0.01, *** *p* < 0.001, and **** *p* < 0.0001. All graphs: ciPTECs *n* = 3. PTECs—proximal tubular epithelial cells, ci-PTECs—conditionally immortalized PTEC, Mac—macrophages, CTRL—control, Cis—cisplatin, and CM—conditioned medium.

**Figure 2 cells-13-00121-f002:**
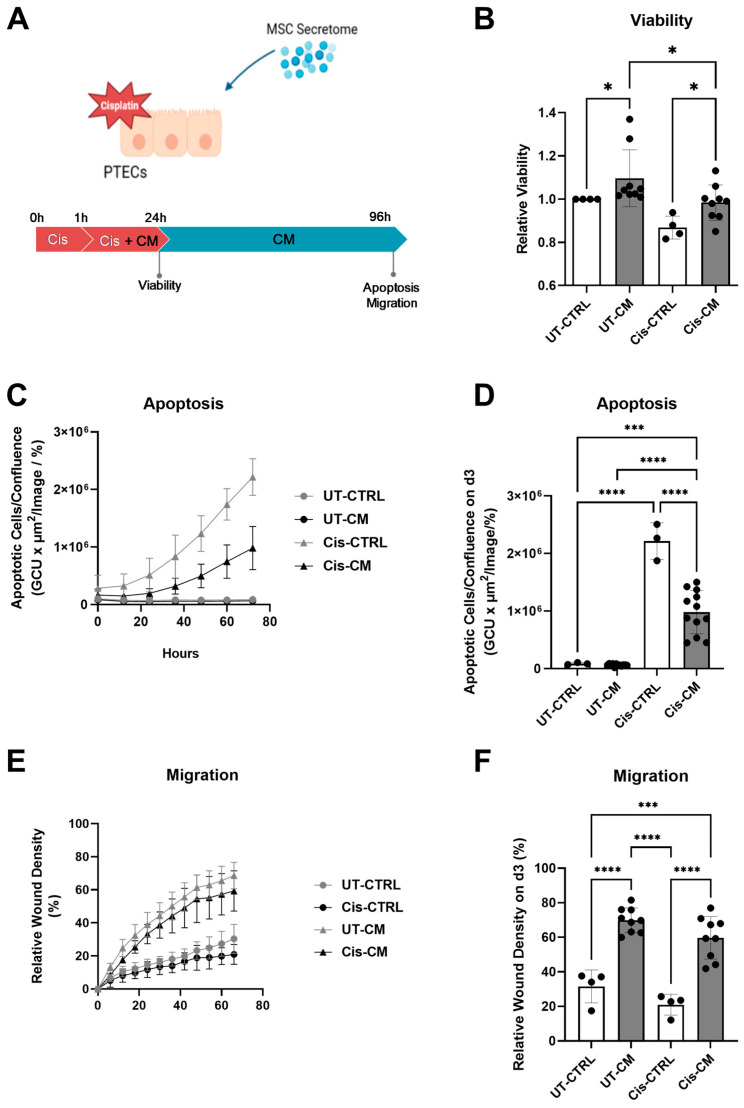
CM attenuated cisplatin-induced cytotoxicity of ciPTECs. CiPTECs were treated with cisplatin for a total of 24 h in the absence and presence of CM (**A**) and the viability was measured using PrestoBlue (**B**). The following day, the cisplatin-containing medium was replaced with either CM or CTRL medium containing Apotracker to monitor the ciPTEC apoptosis for an additional 72 h (**C**). Apoptosis was quantified 96 h post cisplatin treatment and normalized by the cell confluence (**D**). To measure the migratory capacity, the ciPTEC monolayer was scratched after 24 h of cisplatin treatment, the medium was replaced with fresh CM or CTRL medium, and the wound closing was monitored for 3 days (**E**) and quantified 96 h post cisplatin treatment (**F**). One-Way ANOVA with Tukey’s multiple comparisons test was used; significance values: * *p* < 0.05, *** *p* < 0.001, and **** *p* < 0.0001. UT-CTRL and Cis-CTRL n (ciPTECs) = 3; UT-CM and Cis-CM N (ASC) = 3 for viability and 4 for apoptosis assay. (Image in (**A**) created with Biorender). PTECs—proximal epithelial cells, UT-CTRL—untreated control, UT-CM—untreated conditioned medium, Cis-CTRL—cisplatin-treated, Cis-CM—cisplatin and conditioned medium-treated, and GCU—green calibrated unit.

**Figure 3 cells-13-00121-f003:**
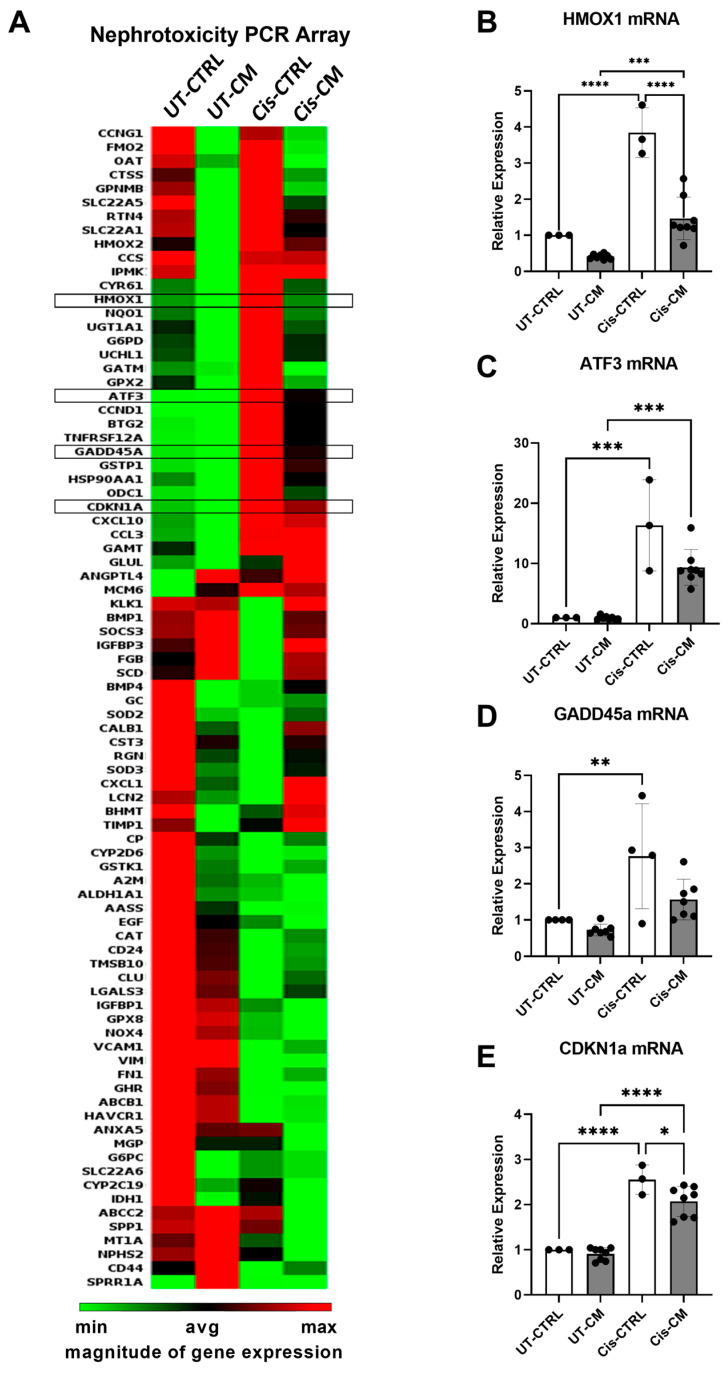
CM attenuated cisplatin-induced expression of nephrotoxicity genes. CiPTECs were treated with cisplatin for a total of 24 h in the absence and presence of CM, then their mRNA was isolated and subjected to a nephrotoxicity PCR array (**A**). Independent validation of array results was performed by RT-qPCR for HMOX-1 (**B**), ATF3 (**C**), GADD45a (**D**), and CDKN1a (**E**). Two-way ANOVA with Tukey’s multiple comparisons test was used; significance values: * *p* < 0.05, ** *p* < 0.01, *** *p* < 0.001, and **** *p* < 0.0001. For PCR array, UT-CTRL and Cis-CTRL n (ciPTECs) = 2; UT-CM and Cis-CM N (ASC) = 4. For RT-qPCR, UT-CTRL and Cis-CTRL n (ciPTECs) = 3; UT-CM and Cis-CM N (ASC) = 4. UT-CTRL—untreated control, UT-CM—untreated conditioned medium, Cis-CTRL—cisplatin-treated, Cis-CM—cisplatin and conditioned medium-treated, HMOX1—heme oxygenase-1, ATF3—Activating Transcription Factor 3, GADD45a—Growth Arrest and DNA Damage-Inducible Alpha, and CDKN1a—cyclin-dependent kinase inhibitor 1A.

**Figure 4 cells-13-00121-f004:**
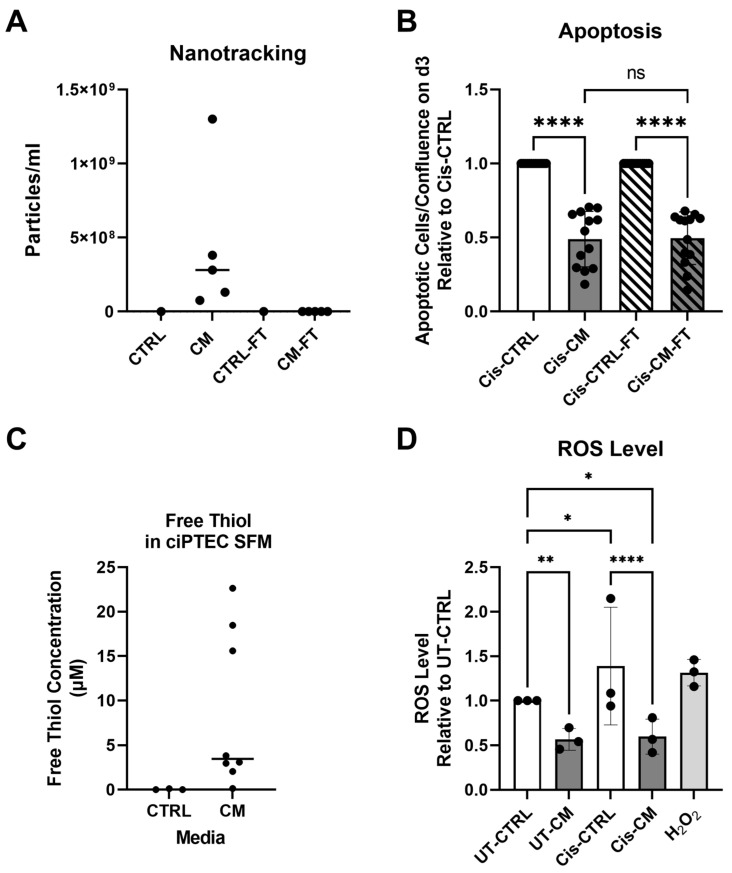
The anti-apoptotic activity of CM was not affected by EV depletion but appeared to be related to its free thiol concentration. The concentration of EVs in CTRL medium and CM with and without EV depletion by 100 kDa ultra-filtration (**A**). The apoptosis level of ciPTECs 3 d after cisplatin removal presented as the relative value of the corresponding Cis-CTRL. (Of note, to compensate for the eventual depletion of soluble factors by ultra-filtration, the apoptotic level of Cis-CM and Cis-CM-FT were normalized to their respective CTRL) (**B**). The concentration of free thiol in the CTRL medium and CM (**C**). The intracellular ROS level of ciPTECs treated with cisplatin and CM; hydrogen peroxide (H_2_O_2_) served as positive control (**D**). Two-way ANOVA with Tukey’s multiple comparisons test was used; significance values: * *p* < 0.05, ** *p* < 0.01, and **** *p* < 0.0001. For NTA and free thiol measurement, CTRL n (ciPTECs) = 3; CM N (ASC) = 4. For apoptosis assay, Cis-CTRL and Cis-CTRL-FT n (ciPTECs) = 4; Cis-CM and Cis-CM-FT N (ASC) = 4. For ROS assay, CT-CTRL, Cis-CTRL, and H_2_O_2_ n (ciPTECs) = 3; UT-CM and Cis-CM N (ASC) = 4. CTRL—control, CM—conditioned medium, CTRL-FT—control medium flow-through, CM-FT—conditioned medium flow-through, UT-CTRL—untreated control, UT-CM—untreated conditioned medium, Cis-CTRL—cisplatin-treated, Cis-CM—cisplatin and conditioned medium-treated, and ROS—reactive oxygen species.

**Figure 5 cells-13-00121-f005:**
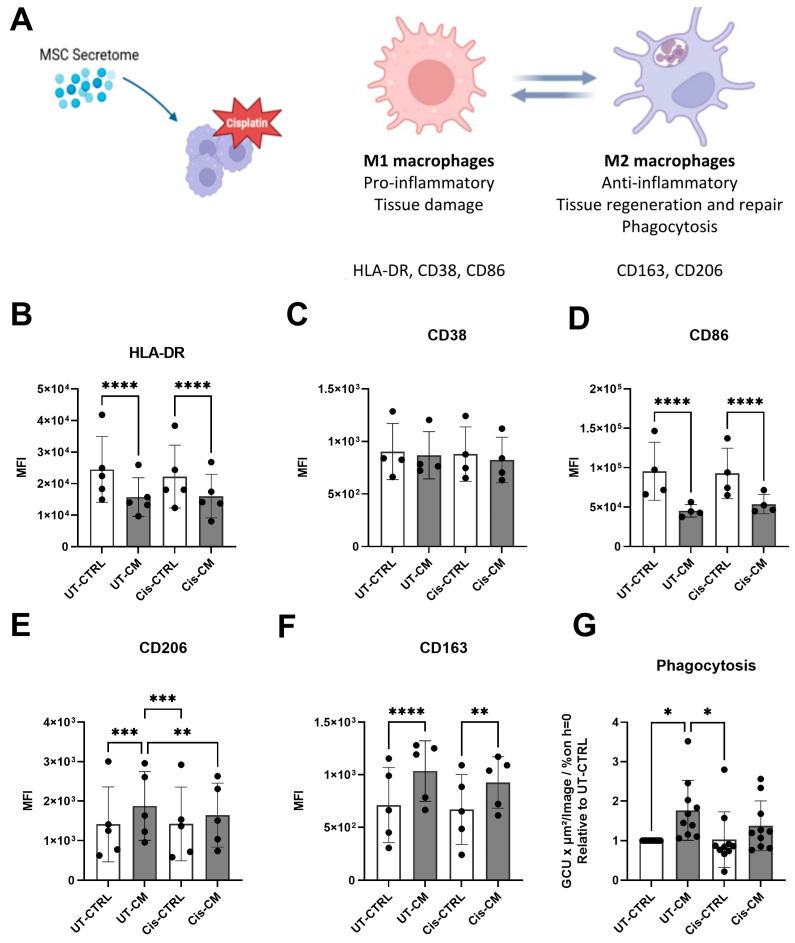
CM skewed macrophage phenotypes towards M2, while cisplatin per se did not affect macrophage phenotype and function. Characteristics of M1 and M2 macrophages relevant for this study (**A**) (Image created with BioRender). Macrophages were treated with cisplatin in the presence and absence of CM for a total of 24 h, then harvested, and their surface markers were measured using FACS. The mean fluorescence intensity (MFI) for HLA-DR (**B**), CD38 (**C**), CD86 (**D**), CD206 (**E**), and CD163 (**F**) is shown. Meanwhile, the phagocytosis capacity of macrophages after the treatment was assessed using *E. coli* bioparticles and monitored for 6 h by live cell imaging. The phagocytosis after 6 h is presented as relative value to that of UT-CTRL macrophages (**G**). Two-way ANOVA with Tukey’s multiple comparisons test was used; significance values: * *p* < 0.05, ** *p* < 0.01, *** *p* < 0.001, and **** *p* < 0.0001. For surface marker measurements, UT-CTRL and Cis-CTRL N (macrophage) = 5; UT-CM and Cis-CM N (ASC) = 4. For phagocytosis assay, UT-CTRL and Cis-CTRL N (macrophage) = 10; UT-CM and Cis-CM N (ASC) = 4. UT-CTRL—untreated control, UT-CM—untreated conditioned medium, Cis-CTRL—cisplatin-treated, Cis-CM—cisplatin and conditioned medium-treated, MFI—mean fluorescence intensity, and GCU—green calibrated units.

**Figure 6 cells-13-00121-f006:**
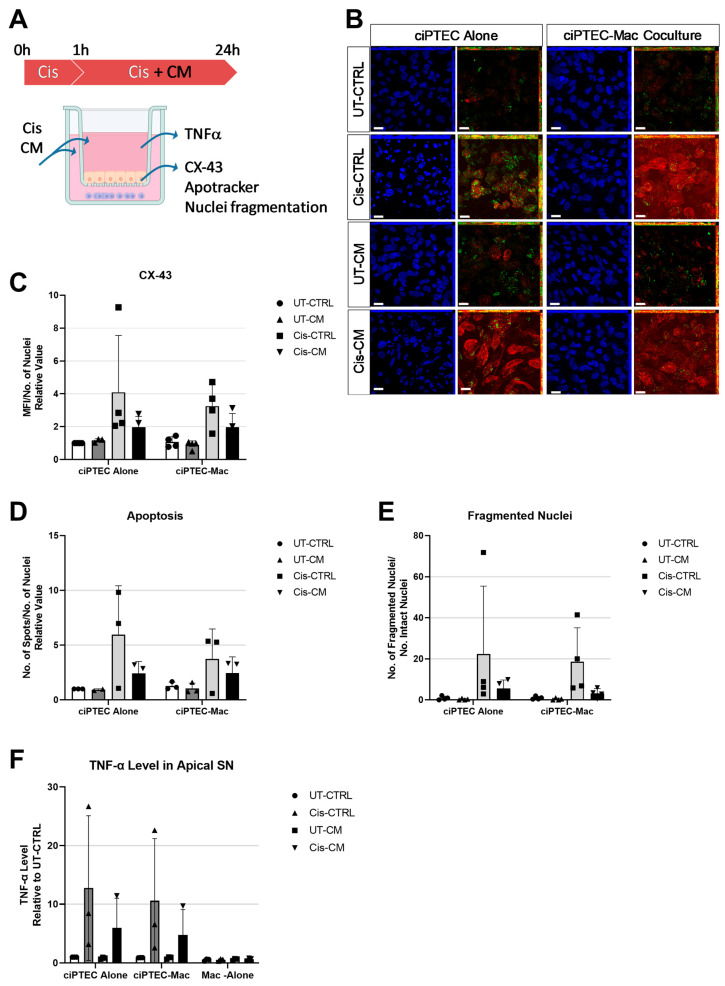
Co-culture with macrophages did not boost the protective effect of CM on ciPTECs. Experimental setup of co-culture system: ciPTECs were seeded on the transwell and macrophages at the basolateral side (**A**) (image created with BioRender). Representative images of ciPTECs on the transwell membrane after being cultured alone or co-cultured with macrophages in the presence or absence of cisplatin with or without CM. Following cisplatin treatment for 24 h, the cells were stained with CX-43 (purple), Apotracker (green), and DAPI (blue) and imaged using confocal microscopy (**B**). Brightness has been adjusted for the CX-43/Apotracker images for better visualization. Scale bars represent 20 µm. The quantification of the fluorescence intensity of CX-43 (**C**), the number of Apotracker spots (**D**), and fragmented nuclei (**E**), normalized by the number of intact nuclei, are presented as the relative value to the UT-CTRL of ciPTEC monoculture (quantification performed on original, non-manipulated images). The level of TNF-α on the supernatant of the apical side of the co-culture after 24 h of cisplatin treatment in the presence or absence of CM, measured by ELISA and presented as the relative value to the UT-CTRL of ciPTEC monoculture (**F**). Three-way ANOVA with Tukey’s multiple comparisons test was used; there were no significant differences. For surface CX-43, fragmented nuclei, and nuclei fragments staining, all groups N (macrophages) = 4. For apoptosis staining and TNF-α ELISA, all groups N (macrophages) = 3 in 3 independent experiments; pooled CM from N = 3 ASC batches each. UT-CTRL—untreated control, UT-CM—untreated conditioned medium, Cis-CTRL—cisplatin-treated, Cis-CM—cisplatin and conditioned medium-treated, TNFα—tumor necrosis factor alpha, MFI—mean fluorescence intensity, ciPTECs—conditionally immortalized proximal tubular epithelial cells, Mac—macrophages, and CX-43—connexin 43.

**Figure 7 cells-13-00121-f007:**
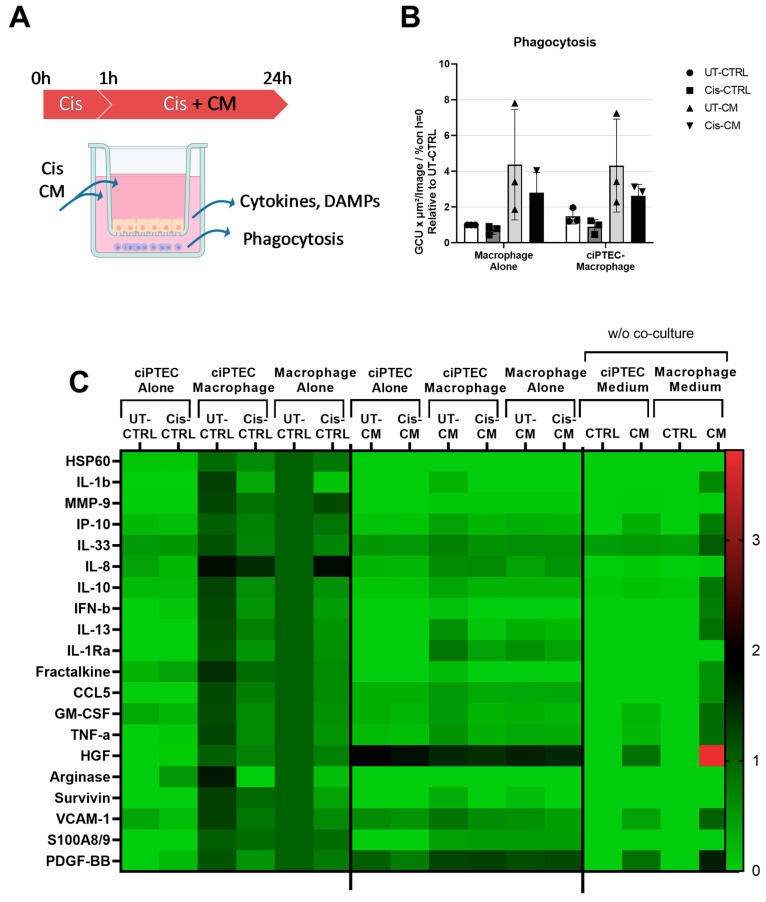
While the co-culture tended to increase macrophage phagocytosis, CM surpassed these effects. Experimental setup (**A**) (image created with BioRender). Phagocytosis tested 24 h post cisplatin treatment presented as the relative value to the UT-CTRL of macrophages cultured alone (**B**). Heat map showing the fold change of analytes measured using a 22-plex panel assay assessing cytokines and DAMPs in the supernatant of basolateral side relative to the value of the UT-CTRL macrophages cultured alone (**C**). Three-way ANOVA with Tukey’s multiple comparisons test was used; the results were non-significant. For phagocytosis and 22-plex panel assay, all groups N (macrophage) = 3 in two independent experiments; pooled CM from N = 3 ASC. UT-CTRL—untreated control, UT-CM—untreated conditioned medium, Cis-CTRL—cisplatin-treated, Cis-CM—cisplatin and conditioned medium-treated, DAMPs—danger-associated molecular patterns, GCU—green calibrated units, w/o—without, HSP—heat shock protein, IL– interleukin, MMP9—matrix metalloproteinase 9, IP-10—interferon-gamma induced protein 10 kD, IFN-b—interferon beta, CCL5—Chemokine (C-C motif) ligand 5, GM-CSF—Granulocyte-macrophage colony-stimulating factor, TNF-a—Tumor necrosis factor-a, HGF—hepatocyte growth factor, VCAM-1—vascular cell adhesion molecule 1, and PDGF-BB—Platelet-derived growth factor subunit B.

**Figure 8 cells-13-00121-f008:**
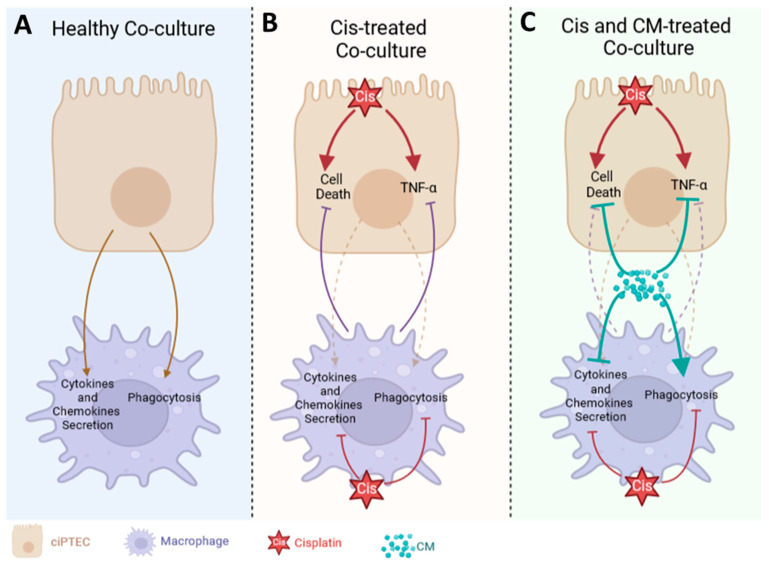
Cisplatin and CM modified the interaction between ciPTECs and macrophages. In healthy co-culture, ciPTECs stimulated macrophages to secrete various cytokines and chemokines and enhanced their phagocytosis (**A**). ciPTEC stimulation on macrophages was later attenuated by the addition of cisplatin which suppressed macrophage secretion and phagocytosis and induced ciPTEC death and TNF-α release. In this setting, macrophages slightly suppressed cisplatin-induced ciPTEC death and TNF-α production (**B**). CM was able to further decrease ciPTEC death and TNF-α release which overruled the regulatory effect of macrophages. In contrast to both cisplatin and ciPTECs, CM augmented macrophage phagocytosis to a greater extent while also concomitantly tuning down cytokines and chemokines secretion (**C**). This image was created using BioRender.

## Data Availability

The datasets used and/or analyzed during the current study are available from the corresponding author on reasonable request.

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
