# Peer review of "Adipose Stromal Cell-Derived Secretome Attenuates Cisplatin-Induced Injury In Vitro Surpassing the Intricate Interplay between Proximal Tubular Epithelial Cells and Macrophages"

_cells, 2024, doi:10.3390/cells13020121_

Round 1
Reviewer 1 Report
Comments and Suggestions for Authors
Rendra et al shows that adipose stromal cells-derived secretome attenuates cisplatin- 2 induced injury in vitro surpassing the intricate interplay be- tween proximal tubular epithelial cells and macrophages.
The study is well designed and data showed are coherent.
I believe that data graph presentation can be improved, not easy for the reader to identify the different groups.
Figure 6B is not good, if possible increase the quality of the image.
The authors suggested MSC CM to decrease cisplatin nephrotoxicity in vivo, it would be good to show that in mice model.
Author Response
Rendra et al shows that adipose stromal cells-derived secretome attenuates cisplatin- 2 induced injury in vitro surpassing the intricate interplay be- tween proximal tubular epithelial cells and macrophages.
The study is well designed and data showed are coherent.
I believe that data graph presentation can be improved, not easy for the reader to identify the different groups.
Figure 6B is not good, if possible increase the quality of the image.
Response: Figure quality appears to have suffered from the pdf conversion during submission. Original tiff and pdf files are of better quality. We adjusted Figure 6B regarding brightness and contrast of the CX-43 and Apotracker images- all graphs adjusted to the same brightness level to allow comparison. The accompanying quantification is based on the original, non-manipulated images.
The authors suggested MSC CM to decrease cisplatin nephrotoxicity in vivo, it would be good to show that in mice model.
Response: We have recently shown that MSCs exert therapeutic effects in a model of cystic kidney disease addressing PKD/Mhm (Cy/+) rats. In the study, we compared ASCs, human skin-derived ABCB5+ stromal cells and their conditioned medium. We observed a moderate improvement of renal function, stronger in the cell-treated animals, but still promising upon conditioned medium infusion (Nardozi et al. J Clin Med . 2022 May 5;11(9):2601. doi: 10.3390/jcm11092601). Furthermore, our manuscript regarding the cisplatin-induced injury model has been accepted just recently. There we report, that clinical-grade human skin-derived ABCB5+ mesenchymal stromal cells exerted anti-apoptotic and anti-inflammatory effects in vitro and modulated mRNA expression towards an anti-inflammatory and pro-regenerative state despite an apparent lack of amelioration of renal damage at physiologic, metabolic, and histologic levels (Rendra et al. Front. Immunol. Sec. Cancer Immunity and Immunotherapy Volume 14 - 2023 | doi: 10.3389/fimmu.2023.1228928

Reviewer 2 Report
Comments and Suggestions for Authors
In the manuscript entitled “Adipose stromal cells-derived secretome attenuates cisplatin-2 induced injury in vitro surpassing the intricate interplay be-3 tween proximal tubular epithelial cells and macrophages” the authors described how the secretome from adipose-derived mesenchymal stem cells attenuates cisplatin derived AKI via reducing apoptosis of proximal tubular epithelial cells (PTECs), and by regulating the PTECs/ macrophage crosstalk.
Overall I believe the manuscript is interesting, and the findings are quite novel for the field. It might shed light on the possible treatment for cisplatin-derived AKI development, although a few experiments and some clarifications need to be addressed to have the manuscript accepted for potential publication in Cells.
Major revisions:
1- In Figure 1 the authors show the effect of a dose-dependent cisplatin treatment on the viability and migration of PTECs. They chose 15uM as concentration for the next set of experiments based on the strong effect on the migratory ability of such cells. However, to have a global idea of the effect of 15uM cisplatin treatment, the authors should also evaluate apoptosis along with cell viability and metabolic activity.
2- From Figure 2 I started to have some issues related to the bar graphs. I have noticed an inconsistency between the number of biological replicates shown in the graphs, and the numbers declared in the legend of each figure. I make an example: in Figure 2B UT-CTRL and Cis-CTRL show 4 black dots, that I assume are the number of biological replicates. However, in UT-CM and Cis-CM I counted many more black dots, 9 to be precise. At first, I thought: How do you compare samples with a big difference in the number of biological replicates? But then I looked into the figure legend and found that the N for each sample =3. It is not clear to me this inconsistency, I am sure the authors have an explanation for it. It is important to point out that this type of inconsistency is present in many figures. Please check
To note, *p < 0.5 I think is incorrect, should be p< 0.05 I assume. The typo is all over the figure legends of the manuscript.
3- Figure 3A, the heatmap nicely shows the PCR array performed on PTECs treated or not with cisplatin and CM from MSCs. The results are quite nice, however, I would emphasize them more in the results section. Figure 3B, 3C and 3D have the same problem brought up in the major revision above (2).
4- In Figure 4 the issue is always the same, please check the paragraphs above.
5- In Figures 5E and 5G I noticed that the statistical analysis of Cis-CTRL vs Cis-CM is missing. Is there a reason for it? I think it is important to have these analyses done.
6- Line 494-496, it is not clear to me what the authors meant, please rephrase to have the reader on the same page with the authors.
7- Figure 6B, the immunofluorescence images have to be improved in quality, unfortunately, they are not as clear as they should be. Figure 6F, I think that the authors should change the scale of the y-axis, to appreciate the results better.
8- Figure 7C is not exhaustive of what the authors claim in the results section. They should represent bar graphs to better compare the levels of the analytes described for each experimental group.
Minor revision:
I would improve the English quality, make it a little more formal. Sometimes I had the feeling it was not as formal as required for a scientific manuscript.
Comments on the Quality of English LanguageI would improve the English quality, make it a little more formal. Sometimes I had the feeling it was not as formal as required for a scientific manuscript.
Author Response
- In Figure 1 the authors show the effect of a dose-dependent cisplatin treatment on the viability and migration of PTECs. They chose 15uM as concentration for the next set of experiments based on the strong effect on the migratory ability of such cells. However, to have a global idea of the effect of 15uM cisplatin treatment, the authors should also evaluate apoptosis along with cell viability and metabolic activity.
Response: We verified the dose-response to cisplatin regarding apoptosis of course, yet not in an experiment with different ASC donors and technical replicates, but rather as proof of concept experiment. We thus show the replicate experiments evaluating viability, apoptosis and migration with the chosen dose of 15 µM.
- From Figure 2 I started to have some issues related to the bar graphs. I have noticed an inconsistency between the number of biological replicates shown in the graphs, and the numbers declared in the legend of each figure. I make an example: in Figure 2B UT-CTRL and Cis-CTRL show 4 black dots, that I assume are the number of biological replicates. However, in UT-CM and Cis-CM I counted many more black dots, 9 to be precise. At first, I thought: How do you compare samples with a big difference in the number of biological replicates? But then I looked into the figure legend and found that the N for each sample =3. It is not clear to me this inconsistency, I am sure the authors have an explanation for it. It is important to point out that this type of inconsistency is present in many figures. Please check
To note, *p < 0.5 I think is incorrect, should be p< 0.05 I assume. The typo is all over the figure legends of the manuscript.
Response: We are more than happy to clarify this point: The number of data points are based on technical (n) plus biological replicates (N). Each experiment has been performed at least three times (technical replicates, n), within each experiments different ASC or macrophage donors have been tested. Thus, control and cisplatin values are for example n=3 (representing the technical replicates), but experimental values are 3 (technical) x 3 (biological replicates) = 9.
Thank you very much for pointing out this important typo. We have changed it of course to p< 0.05
- Figure 3A, the heatmap nicely shows the PCR array performed on PTECs treated or not with cisplatin and CM from MSCs. The results are quite nice, however, I would emphasize them more in the results section. Figure 3B, 3C and 3D have the same problem brought up in the major revision above (2).
Response: We thank the reviewer. We extended the results section to emphasize more the different genes classes modulated by cisplatin and ASC CM and also added information on genes, already affected by CM treatment alone, thus, independent of cisplatin treatment.
- In Figure 4 the issue is always the same, please check the paragraphs above.
- In Figures 5E and 5G I noticed that the statistical analysis of Cis-CTRL vs Cis-CM is missing. Is there a reason for it? I think it is important to have these analyses done.
Response: Thank you very much for this extremely careful evaluation. In fact, we accidentally deleted the significance indicator in Fig. 5E. The statistical analysis for 5G gives a p-value of 0.55, thus not significant.
- Line 494-496, it is not clear to me what the authors meant, please rephrase to have the reader on the same page with the authors.
Response: We have rephrased the sentences and hope, the text is better understandable right now.
- Figure 6B, the immunofluorescence images have to be improved in quality, unfortunately, they are not as clear as they should be. Figure 6F, I think that the authors should change the scale of the y-axis, to appreciate the results better.
Response: We adjusted Figure 6B regarding brightness and contrast of the CX-43 and Apotracker images- all graphs adjusted to the same brightness level to allow comparison. The accompanying quantification is based on the original, non-manipulated images.
Again, we would like to thank the reviewer for to this careful evaluation: we changed the axis of figure.
- Figure 7C is not exhaustive of what the authors claim in the results section. They should represent bar graphs to better compare the levels of the analytes described for each experimental group.
Response: We have added the data as requested as supplementary file showing bar graphs of all the data sets individually. Of note, only the most relevant statistical analyses are shown in these graphs.
Minor revision:
I would improve the English quality, make it a little more formal. Sometimes I had the feeling it was not as formal as required for a scientific manuscript.
Response: The manuscript has been read by a native speaker and she suggested to change some sentences from passive to active voice. We now changed this back as we assume the reviewer refers to these sections. But, when we talk about hypotheses and interpretations, we kept active voice.

Round 2
Reviewer 2 Report
Comments and Suggestions for Authors
Thank you for editing the manuscript according to some of my suggestions. I believe the paper has now improved in several parts and it is now suitable for publication in Cells